# Characterization and Immunological Activity of Exopolysaccharide from *Lacticaseibacillus paracasei* GL1 Isolated from Tibetan Kefir Grains

**DOI:** 10.3390/foods11213330

**Published:** 2022-10-23

**Authors:** Xiaomeng Wang, Juanjuan Tian, Xueliang Zhang, Nanyu Tang, Xin Rui, Qiuqin Zhang, Mingsheng Dong, Wei Li

**Affiliations:** College of Food Science and Technology, Nanjing Agricultural University, Weigang Road, Nanjing 210095, China

**Keywords:** *Lacticaseibacillus paracasei* GL1, exopolysaccharide (EPS), structure characterization, immunomodulatory activity

## Abstract

Two exopolysaccharide fractions (GL1-E1 and GL1-E2) of *Lacticaseibacillus paracasei* GL1 were isolated with the molecular weights of 3.9 × 10^5^ Da and 8.2 × 10^5^ Da, respectively. Both fractions possessed mannose, glucose, and galactose in molar ratios of 1.16:1.00:0.1, and 3.81:1.00:0.12, respectively. A structural arrangement of two fractions was proposed by methylation, one-dimensional and two-dimensional nuclear magnetic resonance experiments. The backbone of GL1-E1 consisted of →4)-*α*-D-Glc*p*(1→, →3,4)-*α*-D-Man*p*(1→, →3,6)-*α*-D-Man*p*(1→, →6)-*α*-D-Man*p*(1→, and →6)-*α*-D-Gal*p*(1→ with *α*-D-Glc*p* at branching point. The backbone of GL1-E2 consisted of →4)-*α*-D-Glc*p*(1→, →3,4)-α-D-Man*p*(1→, →3,6)-α-D-Man*p*(1→, →6)-α-D-Man*p*(1→, →6)-α-D-Gal*p*(1→, and →4)-*β*-D-Man*p*(1→, and the side chain also consisted of *α*-D-Man*p* residue. In addition, the differential scanning calorimetry (DSC) analysis indicated that both GL1-E1 and GL1-E2 had good thermal stability. Furthermore, the two fractions could promote the viability of RAW264.7 cells and exert an immunomodulatory role by enhancing phagocytosis, increasing nitric oxide (NO) release and promoting the expression of cytokines.

## 1. Introduction

Kefir is a traditional beverage fermented by unique probiotic-containing gelatinous particles known as kefir grains. Kefir grains have a complex ecosystem of symbiotic microorganisms with lactic acid bacteria (LAB) and yeasts contained within an exopolysaccharide (EPS) and protein complex [1]. The microbial community composition of kefir grains depends on culture conditions and treatment processing. Rich LAB including *Lactobacillus kefiri*, *L. kefiranofaciens*, *Lacticaseibacillus paracasei*, and *L. helveticus* et al. [2,3] dominate the population in kefir grains. Basic research and clinical treatment have shown the great health benefits of kefir including antitumor [4], immunomodulatory [5], wound healing [6], and anti-allergenic effects [7]. The functional activity of kefir is assigned to microorganisms and the metabolites they produce such as EPS during the growth process.

LAB are generally considered as safe (GRAS) microorganisms, and the EPS synthesized by LAB has a wide structural diversity [8]. Due to the functional characteristics of high-molecular polymers, the production of EPS by LAB has gained widespread interest in the past decades. The beneficial effects of these biopolymers also include anti-tumor activity, cholesterol-lowering ability, anti-hypertensive activity, protection of epithelial cells from intestinal pathogenic microorganisms, and the regulation of fecal microbiota [9]. Among them, a distinctive feature of certain EPS is that they can regulate the host’s immune reaction, or stimulate it to raise the barrier against pathogens. Macrophages not only stimulate the immune system, but also play a key role in the inflammatory response through the release of cytokines. Studies have shown that many polysaccharides isolated from bacteria are involved in immune-stimulating responses, and there is evidence in some studies that many polysaccharides generated of LAB are related to immunostimulatory effects [10]. 

In previous research, most studies have concentrated on the structural localization and functional activities of EPS produced by *L. kefiranofaciens* [11], *Lactiplantibacillus plantarum* [12], and *L. kefiri* [13] in kefir. However, few studies have focused on the structural characteristic and immunomodulatory activity of EPS produced by *L. paracasei* from kefir grains. Among the EPS-producing *Lactobacillus* strains, as a probiotic dietary supplement, *L. paracasei* as a probiotic is recognized for its outstanding probiotic properties and has recently received great attention. Too many probiotic functions have been assigned to *L. paracasei* including the treatment of colitis, interfering with T-cell-driven immune responses [14], and the inhibition of the growth of pathogenic bacteria [15]. At present, the research on the exopolysaccharides produced by *L. paracasei* has mainly focused on the optimization of the culture medium [16,17], the intestinal probiotic function in vitro [18], and the determination of synthetic pathways [19]. 

Recently, the microbial diversity of two kefir grains was analyzed by high-throughput sequencing technology, and GL1 was screened and identified as *L. paracasei* by 16S rDNA sequencing [20]. Although the high-yielding biofilm and EPS-producing strain in kefir grains have been previously determined, the structure and biological functions of the EPS produced by *L. paracasei* GL1 are still unknown. The aim of this study was to research the structure and immune activity of the EPS of *L. paracasei* isolated from Tibetan kefir grains to explore the relationship between the polysaccharide activity and structure. Therefore, we performed the extraction, purification, and structural characterization of EPS from *L. paracasei* GL1 (GL1-EPS) by ultraviolet (UV), high performance liquid chromatography (HPLC), Fourier transform infrared (FT-IR) spectroscopy, gas chromatography-mass spectrometer (GC-MS) as well as one-dimensional (1D) and two-dimensional (2D) nuclear magnetic resonance (NMR). In addition, the immunomodulatory activity of EPS to RAW 264.7 cells induced by lipopolysaccharide (LPS) were determined by examining the cytokine production.

## 2. Materials and Methods

### 2.1. Microorganism and Chemicals

The strain of GL1 was isolated by the traditional streaking separation method from a traditional region of Tibet in Naqu, China, and it was recognized by 16S rRNA sequence analysis. A fragment 16S rDNA of GL1 was amplified by the 16S rDNA primer (Appendix A). The amplification was performed in 50 μL (final volume) of the reaction mixture containing 25 μL Rapid Taq Master Mix, 2 uL of each primer, 8 to 10 ng of purified genomic DNA 2 μL, and 19 μL deionized water. The PCR parameters were 94 °C for 2 min, 30 cycles of 94 °C for 30 s, 55 °C for 30 s, and 72 °C for 90 s, and a final extension at 72 °C for 10 min. Then, the sequences were determined by the BLAST program of the GenBank database, and GL1 was identified as *L. paracasei*. The RAW264.7 murine macrophage cell line was purchased from Type Culture Collection of the Chinese Academy of Sciences, Shanghai, China. MD34 dialysis bags were purchased from Biosharp (Beijing Labgic Technology Co. Ltd., Beijing, China). The nitric oxide (NO) kit was purchased from Nanjing Jiancheng Bioengineering Institute (Nanjing, China). The primers of TNF-α, IL-1β, iNOS, and ꞵ-actin were synthesized by the Sangon Bioengineering Institute (Shanghai, China).

### 2.2. Determination of pH and Bacterial Growth

Cultivations of *L. paracasei* GL1 were performed in De Man Rogosa Sharpe (MRS) medium at 37 °C for 48 h with the inoculum OD_600nm_ value of 1.81 ± 0.021. The pH of different fermented cultures was accomplished by a pH meter. The growth of *L. paracasei* GL1 was characterized by the OD_600nm_ value and the number of viable bacteria. The viable bacteria were determined using the plate count method, and the concentration of the agar power was 2% (*w/v*). Briefly, the fresh culture (1 mL) was mixed with 0.9% (*w/v*) normal saline (9 mL) before making a dilution ranging from 10^−1^ to 10^−6^. The diluted solutions (100 μL) were individually coated on the plates, which were incubated at 37 °C for 48 h. A single colony was counted only with 30–300 colonies [21,22].

### 2.3. Extraction of Crude GL1-EPS

The GL1 strain was inoculated in MRS medium with 4% (*v/v*) inoculum at 37 °C for 48 h, and GL1-EPS was extracted [23]. Briefly, the fermentation was centrifuged at 12,000 rpm for 8 min. Then, the final concentration of 4% (*v/v*) trichloroacetic acid solution was added to the supernatant. After removing the protein precipitation, the culture supernatant was precipitated by adding three-fold ethanol for 12 h at 4 °C. EPS was obtained by centrifugation. After dialyzing the EPS at 4 °C for 72 h, it was freeze-dried to obtain GL1-EPS. The crude GL1-EPS was purified by DEAE-Sepharose Fast Flow and stepwise at a 1 mL/min flow rate with gradient NaCl solution (0, 0.1, and 0.3 M) [24]. Each tube of 10 mL EPS was collected, and the sugar content of GL1-EPS was measured using the phenol-sulfuric acid method [25]. To obtain purified fractions, EPS with different elution concentrations were collected, dialyzed, and lyophilized. The protein content was assessed using the Coomassie brilliant blue method, and the uronic acid content was performed using the m-hydroxybiphenyl method.

### 2.4. Chemical Characterization of GL1-EPS

#### 2.4.1. UV–Vis Analysis and FT-IR Analysis

The UV spectra of 2 mg/mL GL1-E1 and GL1-E2 were recorded by a UV-1603 spectrophotometer (Shimadzu Co., Kyoto, Japan) in the range of 190–500 nm. The FTIR spectra were carried out on a Bruker Tensor-27 FTIR spectrophotometer (Bruker Corp., Billerica, MA, USA) in the frequency range of 4000–400 cm^−1^ [26].

#### 2.4.2. Measurement of Molecular Weight (Mw)

The Mw of GL1-E1 and GL1-E2 was detected by HPLC with a TSK GEL G4000 PWXL column (300 mm × 7.8 mm, Tosoh Corp., Tokyo, Japan). A total of 20 μL of the EPS (2 mg/mL) was injected, and the column was eluted with deionized water at a flow rate of 0.8 mL/min. The standard curve was created by T-Series dextran standards [24].

#### 2.4.3. Monosaccharide Composition Analysis

A total of 5 mg of the samples was added in 2 mL trifluoroacetic acid (2 M) at 120 °C for 2 h. The residual trifluoroacetic acid was removing with methanol by rotary evaporation. Then, a 400 μL hydrolysis sample (dissolved in deionized water), 400 μL 3-methyl-1-phenyl-2-pyrazolin-5-one (PMP) solution (0.5 M), and 200 μL NaOH solution (0.3 M) were mixed at 70 °C for 30 min. After the mixture solution was cooled, it was neutralized with 200 μL HCl solution (0.3 M). Subsequently, the PMP derivatives were vortexed with 4 mL chloroform, and the aqueous phase was centrifuged and filtered to obtain derivatized samples. Finally, a Waters 2695 HPLC system (Milford, MA, America) with a C_18_ column was used to detect the monosaccharide composition at a flow rate of 0.8 mL/min. The mobile phase consists of a solution of 81 mL ammonium acetate (0.1 M), 17 mL acetonitrile, and 2 mL tetrahydrofuran [27].

#### 2.4.4. Methylation Analysis

The methylation analysis of GL1-E1 and GL1-E2 was performed as per a previously described method [28]. The methylation operation was checked by FTIR spectroscopy to verify that the reaction was complete. Briefly, 3 mg of the fully methylated sample was hydrolyzed using 2 mL trifluoroacetic acid (2 M) at 120 °C for 2 h. Subsequently, the hydrolyzed samples were reduced at room temperature for 2 h using NaBH_4_, and 0.1 M acetic acid solution was added to adjust the pH to 5.5. After the solution dried, the solution of pyridine (400 μL) and acetic anhydride (400 μL) was added, and it was acetylated at 100 °C for 1 h. Finally, GL1-E1 and GL1-E2 were analyzed by Thermo Scientific™ TSQ™ 9000 GC-MS (Thermo Fisher Scientific, Inc., Waltham, MA, USA).

#### 2.4.5. 1D and 2D NMR Spectra Analysis

The NMR spectra of samples dissolved in D_2_O were analyzed on a Bruker AVANCE AV-500 spectrometer (Bruker Group, Fällanden, Switzerland) at 500 MHz with a temperature of 313 K. The 2D NMR spectra including the homonuclear correlation spectroscopy (COSY), heteronuclear single quantum coherence edited spectroscopy (HSQC), and nuclear Overhauser effect spectroscopy (NOESY) measurements were recorded [29].

#### 2.4.6. SEM Analysis

The morphology characteristics of the GL1-E1 and GL1-E2 were observed using a SEM (HITACHI S-3000N, Hitachi, Tokyo, Japan). The well-dried samples were treated in an appropriate size and plated on a conductive film. The morphological characteristics of samples were observed at different magnifications [30].

#### 2.4.7. Differential Scanning Calorimetry (DSC) Analysis

The thermal stability analysis of EPS was determined by DSC (NETZSCH-Gerätebau GmbH, Selb, Germany) according to the method of Tian et al. [24]. A sample of 4 mg polysaccharide was accurately weighed and added to the sample tray. During the analysis, the temperature range was set to 25–550 °C with a 10 °C /min heating rate.

### 2.5. Immunomodulatory Effects of GL1-EPS

#### 2.5.1. The Viability of RAW 264.7 Cells

The cytotoxicity assay was measured by the 3-(4,5-Dimethylthiazol-2-yl)-2,5-diphenyltetrazolium bromide (MTT) method [31]. Briefly, RAW 264.7 cells were seeded in a 96-well plate for 24 h, and then treated with different concentrations of EPS (0, 50, 100, 200, and 400 μg/mL) for 24 h. The positive control was the cells handled with 1 μg/mL LPS, while the control group was the cells without EPS treatment. After preprocessing, 200 μL of MTT solution (1 mg/mL) was added to each well and incubated at 37 °C for 4 h. Subsequently, the formazan was dissolved in 150 μL of dimethyl sulfoxide (DMSO). After the formazan was fully dissolved, the absorbance (Ab) at 570 nm was measured. The viability of RAW 264.7 was conducted using the following equation:(1)$$Cell\;viability\;\left(\%\right)=Ab_{sample}\;/\;Ab_{control}\;\times \;100$$

#### 2.5.2. Effect of GL1-EPS on Macrophage Phagocytosis

The phagocytic activity of GL1-EPS was carried out by the described method [27]. After preincubating RAW264.7 cells in a 96-well plate for 24 h, 1 μg/mL LPS and different concentrations (0, 50, 100, 200, and 400 μg/mL) of EPS were added to the plate for 24 h. After removing the supernatant, 100 μL neutral red of 0.075% (*w/v*) was added to each well with the incubation condition at 37 °C for 1 h. Subsequently, 100 μL of the cell lysate was added after removing the excess supernatant. Then, the Ab at 540 nm was measured after cell incubation at 25 °C for 2 h. The phagocytosis index is presented by the following equation:(2)$$Phagocytosis\;index=Ab_{sample}\;/\;Ab_{control}$$

#### 2.5.3. Effect of GL1-EPS on NO Production

The RAW 264.7 cells were cultured in a 96-well plate for 24 h. After the treatment with 1 μg/mL LPS and different concentrations (0, 50, 100, 200, and 400 μg/mL) of EPS to the cells for 24 h, the concentration of NO in the supernatant was determined using the NO kit [27].

#### 2.5.4. Determination of TNF-α, IL-1β, and iNOS Production

The levels of TNF-α, IL-1β, and iNOS production were detected by the RT-PCR method [32]. A total of 1 mL cells were cultured into each well of 6-well plates for 24 h. After the treatment of 1 μg/mL LPS and different concentrations of EPS (0, 50, 100, 200, and 400 μg/mL) for 24 h, the total RNA of fresh cells was extracted by the MolPure^®^ TRIeasy™ Plus Total RNA Kit (Shanghai Yisheng Biotechnology Co. Ltd., Shanghai, China). After the concentration and purity were qualified, the total RNA was reversed into cDNA according to the instructions in the RNA reverse transcription kit. Finally, the expression levels of TNF-α, IL-1β, iNOS, and β-actin were measured by a quantitative reverse-transcription PCR instrument with the calculation of the 2^−ΔΔCt^ method. The primer sequences of TNF-α, IL-1β, iNOS, and β-actin of RAW.264.7 cells are shown in Appendix A, and β-actin was used as a reference gene.

### 2.6. Statistical Analysis

All of the experiments were conducted in triplicate and expressed as the mean ± standard deviation using the software of origin 8.0. The results were analyzed using SPSS (version 16.0, SPSS Inc., Chicago, IL, USA) with *p* < 0.05 considered significant.

## 3. Results and Discussion

### 3.1. Bacterial Growth

The pH values and growth states of the strain GL1 including OD_600nm_ values and viable counts are shown in Figure 1. The viable counts of the strain GL1 peaked at 28 h and reached a maximum of 11.27 log CFU/mL. In contrast, the OD_600nm_ values reached a maximum at 36 h. The pH values dropped throughout the fermentation period, reaching a final pH of 3.86 over time for 48 h. The results of the pH and viable bacteria indicated that GL1 possessed a fast growth characteristic.

### 3.2. Extraction of Crude GL1-EPS

The crude GL1-EPS of *L. paracasei* GL1 was obtained by alcohol precipitation and further purified through DEAE-52 cellulose. The crude GL1-EPS was separated, and two single elution peaks (GL1-E1 and GL1-E2) were detected (Figure 2a). Then, the crude fraction was collected and weighed after lyophilization, and the extraction yield of GL1-EPS reached 460.5 mg/L. The carbohydrate content of the purified GL1-E1 and GL1-E2 was 92.43 ± 1.03% and 93.43 ± 2.03%, respectively, and no protein and uronic acid content were detected by the colorimetric method.

### 3.3. UV and FTIR Spectrum Analysis of GL1-EPS

UV detection of GL1-E1 and GL1-E2 showed no absorption peak at 280 nm, indicating that no protein was found in the purified EPS (Figure 2b). As shown in Figure 2c, the functional groups of GL1-E1 and GL1-E2 were characterized by FTIR analysis. The stretching peak at 3370 cm^−1^ indicated many hydroxyl groups [33] present in the GL1-EPS. A major broad absorption in the 2928 and 2938 cm^−1^ regions were related to the C–H stretching [34], and the intense signal in the 1653 and 1652 cm^−1^ regions corresponded to the C–O group stretching vibration [35]. Absorption bands at 1456 and 1419 cm^−1^ were found, suggesting the presence of ester or ether groups. In addition, the bands in the region of 800 to 1150 cm^−1^ were attributed to the vibration of the C–O–C bond. Additionally, the absorption bands examined at 1028 cm^−1^ and 1055 cm^−1^ of GL1-E1 and GL1-E2 usually represent the stretching vibrations of C–O [36].

### 3.4. Mw and Monosaccharide Composition of GL1-EPS

To further confirm the homogeneity of polysaccharides, Mws of the GL1-E1 and GL1-E2 were determined via HPLC with an ELSD detector (Figure 2d). The elution profile of GL1-E1 and GL1-E2 showed single and narrow peaks, indicating that they were pure and homogeneous. The average Mws of GL1-E1 and GL1-E2 were estimated to be 3.9 × 10^5^ and 8.2 × 10^5^ Da, respectively, according to a calibration curve (Log Mw = −2.3483x + 20.258, R^2^ = 0.9965). Previous studies have concluded that the Mw of EPS extracted from LAB ranges from 10^4^ to 10^6^ Da. The EPS produced by *L. paracasei* KL1-Liu was purified by Sepharose CL-6B to obtain EPSa and EPSb, and structural analysis revealed that the Mw of EPSa and EPSb were 4.6 × 10^4^ Da and 2.1 × 10^4^ Da, respectively [37]. Liu et al. reported that the average Mw of the HY EPS produced by *L. plantarum* was estimated at 9.5 × 10^4^ Da [38], and the EPS produced by *Leuconostoc citreum* L3C1E7 had the average Mw of 5.88 × 10^6^ Da [39]. In addition, many studies have shown that the Mw of LAB-EPS exerts significant impacts on their biological properties. High Mw polysaccharides can promote network formation between EPS and proteins, and they have effects on inhibiting tumor cells [40]. In contrast, the EPS with low Mw produced by *L. plantarum* showed a stronger antioxidant capacity than the other *L. plantarum*.

Through a comparison of the retention times of monosaccharide standards, the monosaccharides of GL1-E1 and GL1-E2 after acid hydrolysis were analyzed. The results in Figure 2e revealed that GL1-E1 is a heteropolysaccharide (HePS) that consists of mannose, glucose, and galactose in molar ratios of 1.16:1.00:0.1, respectively, while GL1-E2 consists of mannose, glucose, and galactose with a molar ratio of 3.81:1.00:0.12, respectively. Overall, there is a wide variety of HePS produced by LAB, and they showed a wide range of variations. Li et al. reported that EPS-S11 produced by *L. paracasei* was composed of mannose, glucose, galactose, and glucuronic acid in molar ratios of 0.87:0.92:1.0:0.24, respectively [41]. Two EPS fractions (EPS1, EPS2) produced by *L. paracasei* ZY-1 isolated from Tibetan kefir grains consisted of mannose and glucose with molar ratios of 1.17:1.00 and 5.18:1.00, respectively [19]. The monosaccharide composition of the EPS from *L. paracasei* appeared to be similar, which usually contains different ratios of mannose, glucose, and galactose. Therefore, the composition ratios of different monosaccharides of the GL1-EPS depend on the species, culture condition, and media ingredient.

### 3.5. Methylation Analysis

The glycosidic bond of EPS was indicated by methylation analysis. The types and proportions of glycosidic bonds of GL1-E1 and GL1-E2 are shown in Table 1. By comparing the ion fragmentation characteristics of EPS with the database, the connection pattern of GL1-E1 and GL1-E2 was obtained. It needs to be mentioned that differences in methylation analysis and monosaccharide composition exist. The reason is that during the hydrolysis process, a part of the sample will be lost, so the measured monosaccharide composition data and methylation data will be biased. The presence of 2,3,4,6-tetra-O-methylglucitol, 2,3,6-tri-O-methylglucitol, 2,3,4-tri-O-methylmannitol, 2,3,4-tri-O-methylgalactitol, 2,6-di-O-methylmannitol, and 2,4-di-O-methylmannitol was found in GL1-E1 using GC-MS analysis, which indicated that GL1-E1 had the linkage of D-Glc*p*-(1→, →4)-D-Glc*p*-(1→, →6)-D-Man*p*(1→, →6)-D-Gal*p*-(1→, →3,4)-D-Man*p*(1→, and →3,6)-D-Man*p*-(1→. Likewise, the existence of 2,3,4,6-tetra-O-methylmannitol, 2,3,6-tri-O-methylglucitol, 2,3,6-tri-methylmannitol, 2,3,4-tri-O-methylmannitol, 2,3,4-tri-O-methylgalactitol, 2,6-di-O-methylmannitol, and 2,4-di-O-methylmannitol showed an association of D-Man*p*-(1→, →4)-D-Glc*p*-(1→, →4)-D-Man*p*-(1→, →6)-D-Man*p-*(1→, →6)-D-Gal*p*-(1→, →3,4)-D-Man*p*(1→, and →3,6)-D-Man*p*-(1→ in GL1-E2. It should be noted that the glycosidic bond ratios of the methylation analysis were different from the results of the monosaccharide composition, and the main reason is that some monosaccharides will be lost during the hydrolysis process, thus, resulting in a slight difference. Balzaretti et al. extracted a novel branched hetero-EPS extracted from *L. paracasei* DG, and GC-MS analysis of the alditol acetates of the EPS revealed the presence of 1,2,3,4,5-penta-O-acetyl-L-rhamnitol, 1,2,3,4,5,6-hexa-O-acetyl-D-galactitol, and 2-acetamido-1,3,4,5,6-penta-O-acetyl-2-deoxy-D-galactitol at a ratio of 4:1:1, respectively [18]. Even though EPS was all isolated from *L. paracasei*, the monosaccharide compositions and the type of glycosidic bonds were quite different, which indicate that the difference in the type in glycosidic bond depends on the difference between strains.

### 3.6. 1D and 2D NMR Spectra Analysis

The NMR spectroscopy is the only analytical tool that can comprehensively analyze the molecular structure of carbohydrates, and sufficient information can be provided to analyze all of the structural characteristics of EPS [42]. The detailed NMR analysis revealed that seven anomeric carbon and proton signals exist in GL1-EPS in Figure 3, and the sugar residues are represented by capital letters. As shown in Figure 3a, six clear anomeric proton peaks were assigned at *δ*5.41, 5.31, 5.17, 5.08, 5.00, and 4.93 ppm in the ^1^H NMR spectrum of GL1-E1, suggesting that GL1-E1 contained six residues (named **A**, **B**, **C**, **D**, **E**, and **F**). All residues were determined as the *α*-configuration based on their chemical shifts greater than 5.0. The anomeric C-1 signals were marked at *δ*100.01, 100.59, 102.39, 102.35, 98.52, and 100.37 ppm from the ^13^C NMR spectrum (Figure 3b). The chemical shifts of six protons can be assigned by the COSY spectra, and the correlations between protons and carbons of GL1-EPS were found by HSQC spectra. The C-1 signals at *δ*100.01, 100.59, 102.39, 102.35, 98.52, and 100.37 ppm could be assigned to **A**, **B**, **C**, **D**, **E**, and **F**, respectively. Furthermore, the assignment of chemical shifts to protons and carbons for the six residues is summarized in Table 2. The proportions of glycosidic bonds in the methylation data were almost the same as in the NMR analysis. Thus, according to the type of glycosidic bond, the assignment of shifts, the residues **A**, **B**, **C**, **D**, **E**, and **F** were concluded to be the →4)-*α*-D-Glc*p*(1→, →3,4)-*α*-D-Man*p*(1→, T-α-D-Glc*p*(1→, →3,6)-*α*-D-Man*p*(1→, →6)-*α*-D-Man*p*(1→, and →6)-*α*-D-Gal*p*(1→, respectively. Finally, the linkage of glycosyl residues in GL1-E1 was confirmed by NOESY spectra (Figure 3e). The results showed that the backbone correlation signal including **A** H-1 to **A** H-4, **A** H-1 to **D** H-3, **D** H-1 to **D** H-3, **D** H-1 to **E** H-6, **E** H1 to **E** H6, **E** H-1 to **B** H-4, and **B** H-1 to **F** H-6, while **C** H-1 to **B** H-3 and **C** H-1 to **D** H-6 were connected in the side chain. Additionally, the repeating unit of GL1-E1 is predicted in Figure 3f. As shown in Figure 3g, seven clear chemical shift signals were found at *δ* 5.41, 5.31, 5.16, 5.08, 5.00, 4.93, and 4.64 ppm of GL1-E2. The six signals of **A**, **B**, **G**, **D**, **E**, and **F** were *α*-configurations, while residue **H** was the *β*-configuration due to the shifts. The signal peaks of GL1-E2 of the remaining carbon were assigned by ^13^C NMR and the HSQC spectrum (Figure 3h,j) at *δ*100.01, 100.59, 102.42, 102.35, 98.52, 100.37, and 102.65 ppm, respectively. The shifts of H-1 to H-6 and C-1 to C-6 for residues **A**, **B**, **G**, **D**, E, **F**, and **H** are assigned in Table 2. Based on the monosaccharide ratios, GC-MS analysis, and NMR shift assignment, the residues **A**, **B**, G, **D**, **E**, **F**, and **H** were assigned to the →4)-*α*-D-Glc*p* (1→, →3,4)-*α*-D-Man*p*(1→, T-*α*-D-Man*p*(1→, →3,6)-*α*-D-Man*p*(1→, →6)-*α*-D-Man*p-*(1→, →6)-*α*-D-Gal*p*(1→, and →4)-*β*-D-Man*p*(1→, respectively. The NOESY spectrum (Figure 3k) indicated that the **H** H-1 to **A** H-4, **A** H-1 to **A** H-4, **A** H-1 to **D** H-3, **D** H-1 to **B** H-3, **B** H-1 to **D** H3, **B** H-1 to **F** H-6, **G** H-1 to **D** H-6, and **G** H-1 to **B** H-4 were in agreement with the **H**-(1→4)-**A**, **A**-(1→4)-**A**, **A**-(1→6)-**E**, **E**-(1→3)-**D**, **D**-(1→3)-**B**, **B**-(1→3)-**D**, **B**-(1→6)-**F**, **G**-(1→6)-**D**, and **G**-(1→4)-**B** linkages, respectively. Finally, the repeat unit structure of GL1-E2 is proposed, as shown in Figure 3l.

### 3.7. SEM Analysis

The secondary electron signal imaging was adopted by SEM to observe the morphology characteristics of EPS. The SEM images of GL1-E1 and GL1-E2 showed that they had different characteristic structures (Figure 4). GL1-E1 had flaky and rod-like irregular structures, which accumulated irregularly. However, the surface of GL1-E2 was smooth with some different sizes of holes. Under 5000 times magnification, both GL1-1 and GL1-E2 had a dense structure, however, the surface shrinkage to form protrusions still existed, and the tightness of GL1-E2 was higher than that of GL1-E1. In addition, the different surface structures of GL1-E1 and GL1-E2 may be related to the aggregation and the purification process of EPS.

### 3.8. DSC Analysis

Heating, as a common method of food processing, will destroy a variety of chemical forces and produce endothermic and exothermic peaks. High temperature may destroy the chemical structure of polysaccharides, even change their spatial conformation, and affect the performance of the activity. Therefore, the thermal stability of polysaccharides will greatly affect the application range of polysaccharides. The thermal property characterization of GL1-E1 and GL1-E2 was carried out by DSC analysis in Figure 5. With the temperature gradually increasing, GL1-E1 and GL1-E2 both had an endothermic and exothermic reaction. The peak temperature of the endothermic reaction of GL1-E1 and GL1-E2 appeared at 300.07 and 295.33 °C, respectively, which were mainly due to the evaporation of water in the polysaccharide component. The peak temperature of the exothermic reaction appeared at 338.66 °C and 344.45 °C, respectively, which was mainly due to the thermal decomposition of the polysaccharide component. In addition, the thermal decomposition temperatures of GL1-E1 and GL1-E2 were 504.19 °C and 492.06 °C, respectively in the third phase. Comparison of the degradation temperature with EPS-S11 (300 °C) and KF5 EPS (279.59 °C) produced by *L. paracasei* [41] showed that both GL1-E1 and GL1-E2 have good thermal stabilities and possess great potential for industrial production.

### 3.9. Immunomodulatory Effects of GL1-E1 and GL1-E2

#### 3.9.1. Cytotoxicity Assay of GL1-E1 and GL1-E2 on RAW264.7 Cells

The immunomodulatory activity of EPS is one of the significant biological activities, and it can regulate the immune system through multiple pathways. RAW264.7 cells are widely used in polysaccharide immune activity research and play a unique role not only in initiating the innate immune response, but also in fighting infection and inflammation by increasing proliferation, phagocytosis, NO production, and secretion of cytokines. To determine the effects of EPS on the acquired immunity of cells, the viability of cells in the presence of GL1-E1 and GL1-E2 was measured using the MTT assay (Figure 6a). The results showed that when the concentration of GL1-E1 and GL1-E2 ranged from 50 to 400 μg/mL, the effects on the RAW264.7 cells were non-toxic. For GL1-E1, the viability of cells increased with an increase in the EPS concentration, although the difference was only significant at the concentration of 50 and 200 μg/mL, and the viability was still higher than that of the blank control. For GL1-E2, the viability increased as the concentration increased, and it was significantly higher (*p* < 0.05) than the control group at a concentration of 400 μg/mL. These results reveal that GL1-E1 and GL1-E2 exert no toxicity to RAW264.7 cells, and in fact, they possessed proliferation on RAW264.7 cells. Similarly, You et al. reported that R-5-EPS extracted from *L. helveticus* could promote the proliferation of RAW264.7 cells when the EPS concentration ranged from 50 to 400 μg/mL [43]. In recent studies, Wang et al. found that an EPS with immunomodulatory effect collected from *L. plantarum* JLK0142 exerted no toxicity on RAW264.7 cells when the treatment concentrations ranged from 50 to 1000 μg/mL [44]. 

#### 3.9.2. Effect of GL1-E1 and GL1-E2 on the Phagocytic Capacity of RAW264.7 Cells

Macrophages are highly conserved phagocytic cells produced in the continuous evolution of the body, and phagocytosis of macrophages is a critical barrier to promote the body’s defense. The strength of the phagocytic ability could represent the activity of the macrophages, and it plays a specific and coordinated role in resisting pathogens and maintaining homeostasis. As shown in Figure 6b, the positive group can promote the phagocytic ability of macrophages. The phagocytic ability of the GL1-E1 and GL1-E2 treatment increased in a dose-dependent manner, and the concentrations of GL1-E1 and GL1-E2 ranged from 50 to 400 μg/mL were significantly higher than that of the control group (*p* < 0.05). These results on the phagocytic capacity of GL1-EPS suggested that GL1-E1 and GL1-E2 can promote the phagocytosis of RAW264.7 macrophages. Similar studies have shown that EPS produced by other LAB can also promote the phagocytic ability of macrophages. Lei et al. found the EPS from *L. kefiri* can enhance the phagocytic activity to the level of 175% at the concentration of 500 μg/mL [45]. For the different fractions of GL1-EPS, the phagocytic capacity of GL1-E2 was stronger than that of GL1-E1, which was closely related to its structure.

#### 3.9.3. Effect of GL1-E1 and GL1-E2 on the NO Production of RAW264.7 Cells

In LPS induced macrophages, NO production is considered to be a typical signal of inflammatory response. As shown in Figure 6c, compared to the control group, GL1-E1 and GL1-E2 significantly increased (*p* < 0.05) the production of NO at the concentration of 50−400 μg/mL. The release of NO increased with the increase in the EPS concentration. At the same time, the production of NO of GL1-E2 was higher than GL1-E1, especially in the high dose of 400 μg/mL, and the NO production of GL1-E2 and GL1-E1 was 41.85 ± 1.18 μM and 24.14 ± 0.51 μM, respectively. The reason for the difference between the two fractions might be that there was more mannose than what GL1-E2 had. Studies have shown that mannose in EPS affects the phagocytosis of macrophages by mediating mannose receptors [46]. In this study, compared with the results that EPS333 isolated from a *Streptococcus thermophilus* strain can stimulate RAW 264.7 to release NO [47], both GL1-E1 and GL1-E2 can also promote the release of NO to increase the immunocyte immunological response. Several studies have also shown that EPS can activate the immune response by increasing the release of NO [44].

#### 3.9.4. Effects of GL1-E1 and GL1-E2 on Cytokines Release

Activated macrophages secrete many cytokines that can activate and modulate immune responses, and simultaneously, they can clear pathogens and inhibit cancer cells in some special patterns [48]. Therefore, detection of the expression of three typical cytokines (TNF-α, IL-1ꞵ, and iNOS) is a critical way to evaluate the immunological activities of GL1-E1 and GL1-E2 on macrophages. As shown in Figure 6d–f, macrophages that were not treated with EPS secreted the minimum levels of TNF-α, IL-1β, and iNOS, while the relative expression of TNF-α, IL-1β, and iNOS in RAW264.7 cells increased in a dose-dependent manner. After treatment with 400 μg/mL GL1-E1, the relative expression of TNF-α, IL-1β, and iNOS were 2.69, 2.91, and 1.16 times, respectively, which were significantly higher (*p* < 0.05) than that of the control group. To the high dose concentration of GL1-E2 of 400 μg/mL to RAW264.7 cells, the relative expression of TNF-α, IL-1β, and iNOS was 2.69, 8.27, and 4.11 times, respectively. The results indicate that GL1-E1 and GL1-E2 could enhance the release of cytokines in RAW264.7 cells, suggesting that GL1-E1 and GL1-E2 may play a role in the immune response. Our results were consistent with the results where EPS from *L. plantarum* MM89 could significantly enhance the relative expression of TNF-α and IL-1β [49]. A similar result that EPS1190 from *S. thermophilus* CRL1190 could increase the production of pro-inflammatory cytokines has also been found [13]. Hee et al. isolated the intact bacterial cell *L. paracasei* KB28 attached to EPS from kimchi, and studies have shown that EPS-producing *L. paracasei* KB28 induced the expression of TNF-α, IL-6, and IL-12 in mouse macrophages [50]. Therefore, it is very valuable to study the immunomodulatory effects of EPS produced by *L. paracasei*. 

The immune regulatory response induced by LAB-produced EPS is being investigated. EPS synthesized by *L. casei* acts by reducing immune cell overreaction [51], and the EPS produced by *L. rhamnosus* RW-9595M can induce the low expression levels of TNF-α and IL-6 to macrophages [52]. However, it is impractical to define common immune outcomes for LAB EPS because their extensive structural diversity may affect the recognition of immune system receptors, resulting in differences in immune regulation mechanisms. The immunomodulatory activities of EPS are closely related to structural characteristics including monosaccharide composition ratios, Mw, glycosidic bond, and backbone linkage. The immunomodulatory activity of EPS generally involves different signaling pathways, and the signaling pathways are determined by the binding of cellular receptors to EPS, while cell receptors prefer low Mw EPS [24]. Zhu et al. evaluated EPS0142 from *L. plantarum* RS20D, which consists of glucose, galactose, and glucosamine with a low Mw of 1.69 × 10^6^ Da, which could significantly enhance the immunomodulatory activity of RAW264.7 [53]. In our study, GL1-E2 with a relatively higher Mw showed better immunomodulatory activity. The reason for it being in contrast to EPS0142 may be due to the influence of other factors (monosaccharide composition, glycosidic-linkage, and glycosidic-linkage) being much more than that of Mw, moreover, the Mw of GL1-E1 and GL1-E2 did not differ much. Furthermore, glycosidic bonds can affect the biological activities of EPS. Previous research has shown that EPS composed of *α*-mannose or *α*-glucose may be likely to exhibit strong immunomodulatory activities on macrophages [54], and certain glycosidic bonds in EPS such as *α*-(1→3)-mannose, *α*-(1→4)-glucose, and *α*-(1→6)-glucose are closely related to immune activity [55]. Both GL1-E1 and GL1-E2 possessed glycosidic bonds of *α*-(1→3)-mannose and *α*-(1→4)-glucose, and they exhibited strong immunoregulatory activity. In addition, the differences in the biological activities of polysaccharides were not only affected by one factor, but a composite effect of multiple factors. Hence, a full understanding of the relationship between the structure characteristic and immunoregulatory activity of EPS is quite essential.

## 4. Conclusions

GL1-EPS from *L. paracasei* isolated from Tibetan kefir grains was investigated. The structure characteristics of two purified fractions (GL1-E1 and GL1-E2) were uncovered. 

These were heteropolysaccharides with different Mws and are composed of mannose, glucose, and galactose with different molar ratios. The structural characterization results suggest that both GL1-E1 and GL1-E2 possessed complex structures. Furthermore, GL1-EPS exerted strong immunomodulatory activity by increasing the viability of the RAW264.7 cells, enhancing phagocytosis, and promoting the production of NO and cytokines (TNF-α, IL-1β, and iNOS). To gain more understanding of the immunomodulatory activity, the regulatory roles of GL1-E1 and GL1-E2 on the cell signaling pathways will be the focus of further research.

## Figures and Tables

**Figure 1 foods-11-03330-f001:**
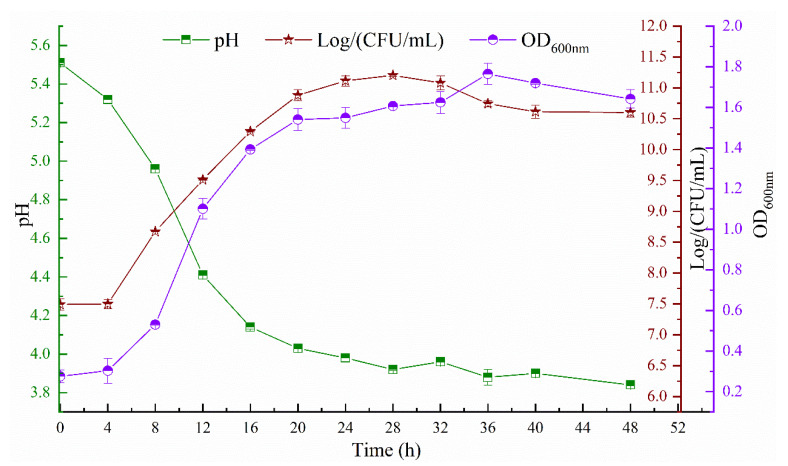
Kinetics curve of bacterial growth and pH values by *L. paracasei* GL1.

**Figure 2 foods-11-03330-f002:**
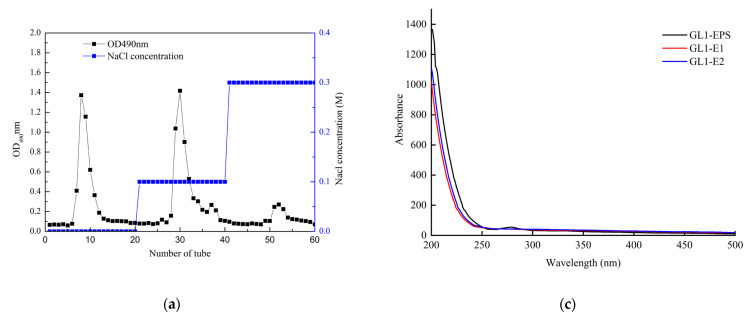
Elution curves of the crude EPS (**a**), UV spectra (**b**), FT-IR spectra (**c**), Mw distributions (**d**), and monosaccharide compositions (**e**) of GL1-E1 and GL1-E2 derived from *L. paracasei* GL1, respectively.

**Figure 3 foods-11-03330-f003:**
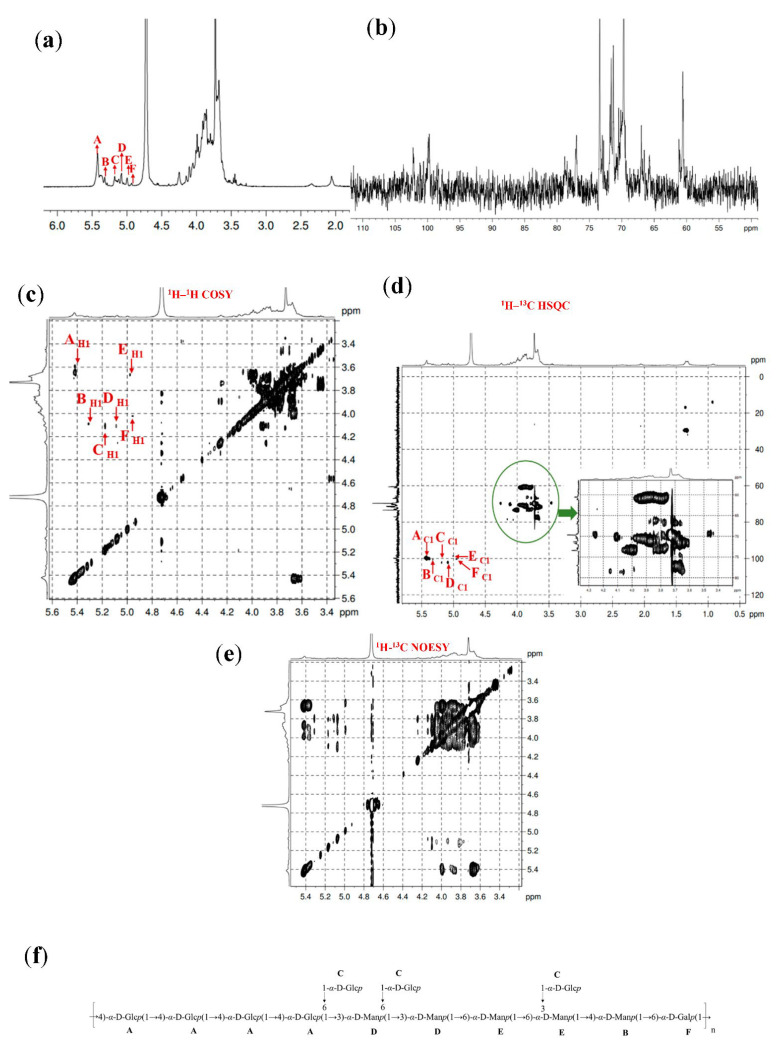
The NMR spectra and proposed structures of GL1-E1 and GL1-E2. The ^1^H NMR spectrum of GL1-E1 (**a**), ^13^C NMR spectrum of GL1-E1 (**b**), ^1^H^1^–H COSY spectrum of GL1-E1 (**c**), ^1^H–^13^C HSQC spectrum of GL1-E1 (**d**), NOESY spectrum of GL1-E1 (**e**), and proposed structure of GL1-E1 (**f**); ^1^H NMR spectrum of GL1-E2 (**g**), ^13^C NMR spectrum of GL1-E2 (**h**), ^1^H–^1^H COSY spectrum of GL1-E2 (**i**), ^1^H–^13^C HSQC spectrum of GL1-E2 (**j**), NOESY spectrum of GL1-E2 (**k**), and proposed structure of GL1-E2 (**l**).

**Figure 4 foods-11-03330-f004:**
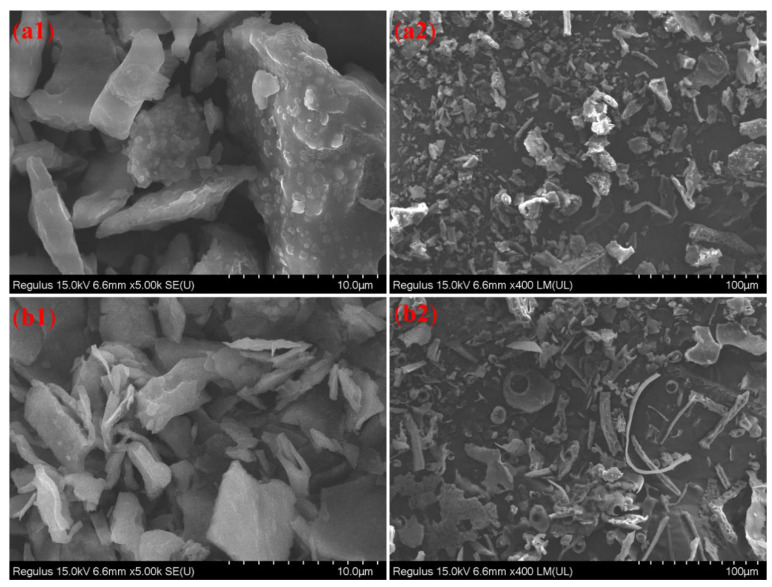
SEM images of GL1-E1 and GL1-E2. GL1-E1 images of different magnifications (**a1**: ×5000; **a2**: ×400) and GL1-E2 images of different magnifications (**b1**: ×5000; **b2**: ×400).

**Figure 5 foods-11-03330-f005:**
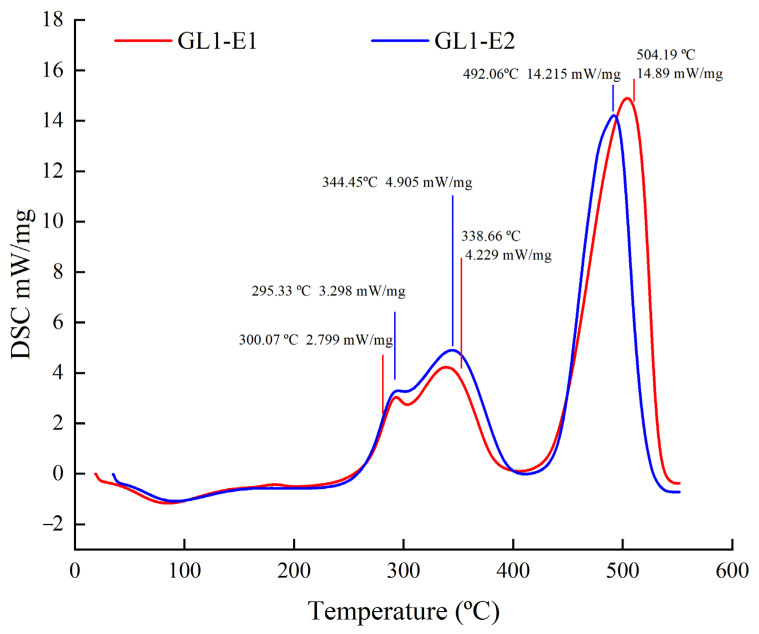
The DSC curves of GL1-E1 and GL1-E2.

**Figure 6 foods-11-03330-f006:**
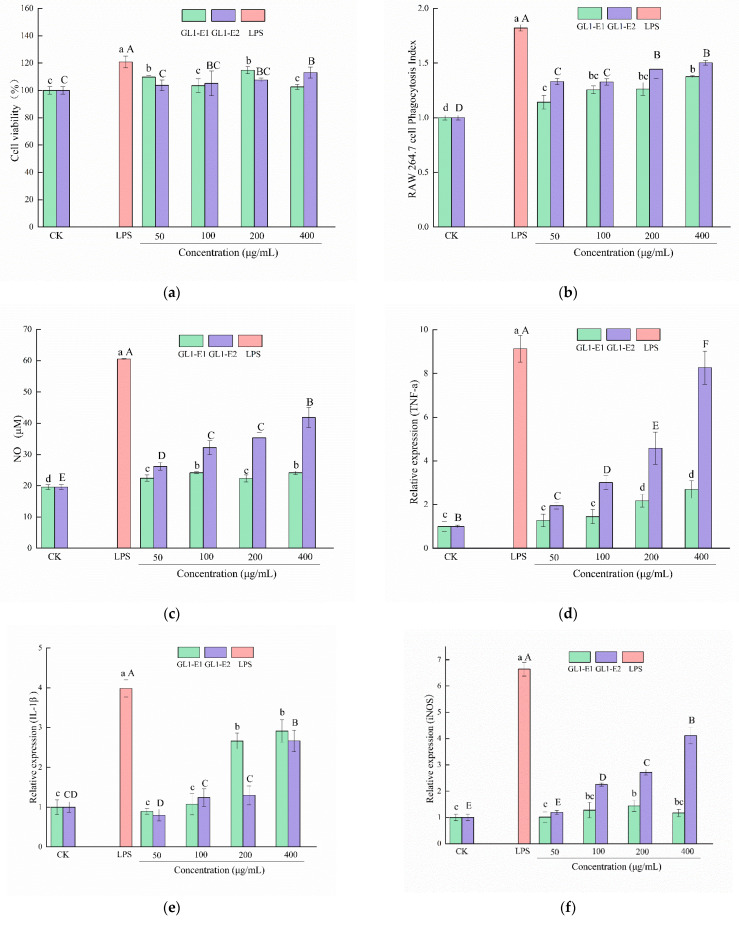
Effects of GL1-E1 and GL1-E2 on RAW264.7 cells in vitro. Cell viability (**a**), phagocytosis index (**b**), NO production (**c**), TNF-α (**d**), IL-1ꞵ (**e**), and iNOS (**f**). Lowercase letters (a–d) and uppercase letters (A–F) indicated significant differences among different concentrations (*p* < 0.05) within the GL1-E1 and GL1-E2 groups, respectively.

**Table 1 foods-11-03330-t001:** Glycosidic linkage analysis of the constituent sugar of GL1-E1 and GL1-E2.

Fractions	Methylation Sugar	Glycosidic Linkages	Mass Fractions (m/z)	Relative Molar Ratios
GL1-E1	2,3,4,6-Me_4_-Glc	D-Glc*p*-(1→	71, 87, 102, 117, 129, 145, 162, 205	2.6
	2,3,6-Me_3_-Glc	→4)-D-Glc*p*-(1→	71, 87, 99, 117, 129, 142, 159, 233	4.1
	2,3,4-Me_3_-Man	→6)-D-Man*p-*(1→	71, 87, 99, 118, 129, 159, 161, 189, 233	2.6
	2,3,4-Me_3_-Gal	→6)-D-Gal*p*-(1→	59, 71, 87, 99, 117, 129, 159, 189	1.0
	2,6-Me_2_-Man	→3,4)-D-Man*p*-(1→	87, 117, 129, 160, 172, 185, 203	1.4
	2,4-Me_2_-Man	→3,6)-D-Man*p*-(1→	59, 87, 101, 117, 129, 189, 233	2.3
GL1-E2	2,3,4,6-Me_4_-Man	D-Man*p*-(1→	71, 87, 102, 117, 129, 145, 162, 205	4.4
	2,3,6-Me_3_-Glc	→4)-D-Glc*p*-(1→	71, 87, 99, 117, 129, 142, 159, 233	3.2
	2,3,6-Me_3_-Man	→4)-D-Man*p*-(1→	71, 87, 102, 117, 129, 143, 162, 189, 233	1.2
	2,3,4-Me_3_-Man	→6)-D-Man*p-*(1→	71, 87, 99, 118, 129, 159, 161, 189, 233	1.0
	2,3,4-Me_3_-Gal	→6)-D-Gal*p*-(1→	59, 71, 87, 99, 117, 129, 159, 189	1.0
	2,6-Me_2_-Man	→3,4)-D-Man*p*-(1→	87, 117, 129, 160, 172, 185, 203	2.2
	2,4-Me_2_-Man	→3,6)-D-Man*p*-(1→	59, 87, 101, 117, 129, 189, 233	2.1

**Table 2 foods-11-03330-t002:** Chemical shifts (ppm) of the ^1^H and ^13^C signals for those recorded in D_2_O at 313 K.

Fractions	Residues	Sugar Linkages	H1/C1	H2/C2	H3/C3	H4/C4	H5/C5	H6/C6
GL1-E1	**A**	→4)-*α*-D-Glc*p-*(1→	5.41	3.64	3.72	3.91	3.69	3.81
			100.01	73.44	73.76	78.23	73.27	60.73
	**B**	→3,4)-*α*-D-Man*p-*(1→	5.31	4.15	3.82	4.01	3.65	3.59
			100.59	69.69	79.18	78.23	72.93	60.90
	**C**	T-*α*-D-Glc*p-*(1→	5.17	4.10	3.91	3.78	3.48	4.00
			102.39	68.84	73.39	71.00	71.49	60.69
	**D**	→3,6)-*α*-D-Man*p-*(1→	5.08	4.10	3.88	3.69	3.76	3.72
			102.35	68.82	78.33	73.76	71.72	70.32
	**E**	→6)-*α*-D-Man*p-*(1→	5.00	3.62	3.98	3.80	3.89	3.66
			98.52	71.18	73.42	73.49	71.11	69.87
	**F**	→6)-*α*-D-Gal*p-*(1→	4.93	4.02	3.85	3.91	3.68	3.81
			100.37	70.37	70.96	70.22	70.68	70.54
GL1-E2	**A**	→4)-*α*-D-Glc*p*-(1→	5.41	3.64	3.72	3.91	3.69	3.81
			100.01	73.44	73.76	78.23	73.27	60.73
	**B**	→3,4)-*α*-D-Man*p*-(1→	5.31	4.15	3.82	4.01	3.65	3.59
			100.59	69.69	79.18	78.23	70.93	60.90
	**G**	T-*α*-D-Man*p-*(1→	5.16	4.11	3.94	3.67	3.79	3.72
			102.42	68.55	73.66	73.22	71.49	61.15
	**D**	→3,6)-*α*-D-Man*p-*(1→	5.08	4.10	3.88	3.69	3.76	3.72
			102.35	68.82	78.33	73.76	71.72	70.32
	**E**	→6)-*α*-D-Man*p-*(1→	5.00	3.62	3.98	3.80	3.89	3.66
			98.52	71.18	73.42	73.49	71.11	69.87
	**F**	→6)-*α*-D-Gal*p-*(1→	4.93	4.02	3.85	3.91	3.68	3.81
			100.37	70.37	70.96	70.22	70.68	70.54
	**H**	→4)-*β*-D-Man*p*-(1→	4.64	3.85	3.69	3.92	3.77	3.61
			102.65	70.99	73.66	78.22	71.38	61.13

## Data Availability

The data presented in this study are available on request from the corresponding author.

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
