# Peer review of "Characterization and Immunological Activity of Exopolysaccharide from Lacticaseibacillus paracasei GL1 Isolated from Tibetan Kefir Grains"

_foods, 2022, doi:10.3390/foods11213330_

Round 1
Reviewer 1 Report
General comments to manuscript Foods-1961877This manuscript entitled “characterization and immunological activity of exopolysaccharide from Lacticaseibacillus paracasei GL1 isolated from Ti-betan kefir grains “ presented the structural characterization of two polysaccharides and were immunological activity
Specific comments:
Introduction:
Line 53: Put a “.” after “in kefir”.
Materiel and methods:
Line 87: no fucose, xylose, arabinose, fructose, galacturonic acid or glucuronic acid were used as standard?
Line 98: what kind of agar was used for the plate count method?
Line 141: “purified fractions”: which ones?
Line 150: “2.4.51.”: pay attention to the numbering.
Line 151: What is the solvent used for the NMR experiments.
Results:
Line 228: “no protein were found in the purified EPS”: has a protein assay by colorimetric method been carried out?
Line 241: “The elution profile of 240 GL1-E1 and GL1-E2 showed single and narrow peaks”: Figure 2d, the GL1-E1 present a small and broad peak at 12 min: is it another part of the polysaccharide?
Line 258: “GL1-E1 was a heteropolysaccharide (HePS), which consists of mannose, glucose, and galactose in molar ratios of 1.16: 1.00: 0.1”: in figure 2e, peaks were also observed at 12 and 20 min: can you assign these peaks to other standard sugar?
Line 259: “GL1-E2 consisted of mannose, glucose, and galactose with a molar ratio of 3.81: 1.00: 0.12” : in figure 2e, peaks were also observed at 12 and two at 20 min: can you assign these peaks to other standard sugar?
Line 304: “3.6.1.”: pay attention to the numbering
Line 309-311: GL1-EPS possess six or seven anomeric carbon and proton signals?
Figure 3a: GL1-E1: one signal was observed à 2 ppm: can you assign it?
In FT-IR analysis, you mentioned the presence of C=O group in EPS. This C=O was not present in NMR spectra?
Line 333-334 and Figure 3h: one signal was observed à 98 ppm: can you assign it?
Figure 3g: GL1-E2: one signal was observed à 2 ppm: can you assign it?
In FT-IR analysis, you mentioned the presence of C=O group in EPS. This C=O was not present in NMR spectra?
Do polysaccharides have substituent groups such as acetate?
Author Response
Reviewer #1 comments:
1 Introduction: Line 53: Put a “.” after “in kefir”.
Response: Thank you for your comment. We have put a “.” after “in kefir” in Line 52.
2 Materiel and methods: Line 87: no fucose, xylose, arabinose, fructose, galacturonic acid or glucuronic acid were used as standard?
Response: Thank you for your question. We used mannose, rhamnose, glucose, galactose, and arabinose as standard. Standard were selected according to the common monosaccharide composition of exopolysaccharides produced by lactic acid bacteria, and after determination, the monosaccharide composition of purified GL1-E1 and GL1-E2 could be corresponding to the standard. Thus, we didn't choose any more monosaccharides as standards.
3 Line 98: what kind of agar was used for the plate count method?
Response: Thank you for your question. We used agar power purchased from Beijing Aoboxing Bio-tech Co., LTD. And the concentration of the agar power was 2% (w/v). We added “and the concentration of the agar power was 2% (w/v).” in Lines 97–98.
4 Line 141: “purified fractions”: which ones?
Response: Thank you for your suggestion. The purified fractions were GL1-E1 and GL1-E2. We added “(GL1-E1 and GL1-E2)” in Line 139.
5 Line 150: “2.4.51.”: pay attention to the numbering.
Response: Thank you for your comment. We checked Line 148 “2.4.5 1D and 2D NMR spectra analysis” and made sure it was correct.
6 Line 151: What is the solvent used for the NMR experiments.
Response: Thank you. The solvent used for the NMR experiments was D2O. And we supplemented “dissolved in D2O” in Line 149.
7 Results: Line 228: “no protein were found in the purified EPS”: has a protein assay by colorimetric method been carried out?
Response: Thank you. We wrote the method of “Protein content was assessed using Coomassie brilliant blue method” in Lines 113–114. We used colorimetric method to carry out the protein content and no protein were found in the purified EPS. We revised “and no protein and uronic acid content were detected.” into “and no protein and uronic acid content were detected by colorimetric method.” in Lines 221–222.
8 Line 241: “The elution profile of 240 GL1-E1 and GL1-E2 showed single and narrow peaks”: Figure 2d, the GL1-E1 present a small and broad peak at 12 min: is it another part of the polysaccharide?
Response: Thank you for your question. We checked and repeatedly confirmed that the small peak at 12 min was an impurity peak. We determined the Mw of GL1-E1 under the same conditions, and finally obtained the HPLC peak. We revised Figure 2d in Line 268.
9 Line 258: “GL1-E1 was a heteropolysaccharide (HePS), which consists of mannose, glucose, and galactose in molar ratios of 1.16: 1.00: 0.1”: in figure 2e, peaks were also observed at 12 and 20 min: can you assign these peaks to other standard sugar?
Response: Thank you for your question. In Figure 2e, peaks observed at 12 and 20 min were impurities from the process of hydrolysis. The reason why there was no impurity peak in the standard curve is that the standard was derived directly without hydrolysis.
10 Line 259: “GL1-E2 consisted of mannose, glucose, and galactose with a molar ratio of 3.81: 1.00: 0.12” : in figure 2e, peaks were also observed at 12 and two at 20 min: can you assign these peaks to other standard sugar?
Response: Thank you for your question. In Figure 2e, peaks observed at 12 and 20 min were impurities from the process of hydrolysis. The impurities peaks were found both in GL1-E1 and GL-E2 samples. The reason why there was no impurity peak in the standard curve is that the standard was derived directly without hydrolysis.
11 Line 304: “3.6.1.”: pay attention to the numbering
Response: Thank you for your comment. We checked Line 304 “3.6. 1D and 2D NMR spectra analysis” and made sure it was correct.
12 Line 309-311: GL1-EPS possess six or seven anomeric carbon and proton signals?
Response: Thank you for your question. As shown in Figure 3a, six clear anomeric proton peaks were assigned at δ5.41, 5.31, 5.17, 5.08, 5.00, and 4.93 ppm in the 1H NMR spectrum of GL1-E1, suggesting that GL1-E1 contained six residues (named A, B, C, D, E, and F). As shown in Figure 3g, seven clear chemical shift signals were found at δ 5.41, 5.31, 5.16, 5.08, 5.00, 4.93, and 4.64 ppm of GL1-E2. GL1-E1 and GL1-E2 had five identical glycosidic bonds and three different glycosidic bonds. So GL1-EPS possess eight anomeric carbon and proton signals.
13 Figure 3a: GL1-E1: one signal was observed à 2 ppm: can you assign it?
Response: Thank you very much for your question. According to your question, we re-analyzed this signal and found that this small peak had no cross link with other signals in 2D NMR. For this reason, this peak is not related to us, so we did not assign the signal observed at 2 ppm.
14 In FT-IR analysis, you mentioned the presence of C=O group in EPS. This C=O was not present in NMR spectra?
Response: Thank you very much for your comment. We are very sorry for the ascription mistake of the intense signal in the 1,653 and 1,652 cm-1 regions of FT-IR spectrum. We revised the sentence into “and the intense signal in the 1,653 and 1,652 cm-1 regions, corresponding to C-O group stretching vibration [35].” in Lines 229–230. GL1-EPS had no C=O, so it was not presented in NMR spectra.
15 Line 333-334 and Figure 3h: one signal was observed à 98 ppm: can you assign it?
Response: Thank you very much for your discovery of our structural problems. The structural analysis of polysaccharides plays an important role in the study of their functional properties. We are very sorry for the missing of the signal observed at 98 ppm. We checked carefully and attributed the signal of GL1-E2 according to the comments made by the reviewer, and 98 ppm signal was assigned again. The results showed that the signal at 98 ppm belonged to →6)-α-D-Manp-(1→. We revised the structure of GL1-E2 in the manuscript and predicted the possible structure of GL1-E2 in Line 359.
16 Figure 3g: GL1-E2: one signal was observed à 2 ppm: can you assign it?
Response: Thank you very much for your question. According to your question, we re-analyzed this signal and found that this small peak had no cross link with other signals in 2D NMR. For this reason, this peak is not related to us, so we did not assign the signal observed at 2 ppm.
17 In FT-IR analysis, you mentioned the presence of C=O group in EPS. This C=O was not present in NMR spectra?
Response: Thank you very much for your comment. We are very sorry for the ascription mistake of the intense signal in the 1,653 and 1,652 cm-1 regions of FT-IR spectrum. We revised the sentence into “and the intense signal in the 1,653 and 1,652 cm-1 regions, corresponding to C-O group stretching vibration [35].” in Lines 229–230. GL1-EPS had no C=O, so it was not presented in NMR spectra.
18 Do polysaccharides have substituent groups such as acetate?
Response: Thank you for your valuable question. Acetate content was determined by HPLC analysis after hydrolysis of the polymer with TFA (2 M for 2 h at 120 °C), and no acetate were detected. We have read a lot of literature and found that the EPS produced by LAB did not contain acetate substituents, and our results were consistent with these results.

Reviewer 2 Report
Dear Editors and authors,
The manuscript (Characterization and immunological activity of exopolysaccha ride from Lacticaseibacillus paracasei GL1 isolated from Tibetan kefir grains) has a good idea but needs some correcting before it can be accepted for publication.
1-The purpose of the manuscript is not clear, a clear aim should be written at the end of the introduction.
2-Scientific names should be reviewed and written according to the modern nomenclature of lactic acid bacteria.
3-Many working methods do not contain scientific references such as
Determination of pH and bacterial growth, UV-vis analysis and FT-IR analysis, D and 2D NMR spectra analysis, SEM analysis, The viability of RAW 264.7 cells, Effect of GL1-EPS on NO production and Determination of TNF-α, IL-1β and iNOS production
I suggest you two reference (Kimoto-Nira, H., Ohashi, Y., Amamiya, M., Moriya, N., Ohmori, H., & Sekiyama, Y. (2020). Fermentation of onion (Allium cepa L.) peel by lactic acid bacteria for production of functional food. Journal of Food Measurement and Characterization, 14(1), 142-149.). Ultrasound treatment (low frequency) effects on probiotic bacteria growth in fermented milk. Future of Food: Journal on Food, Agriculture and Society, 7(2), Nr-103.)) for Determination of pH and bacterial growth and (Li, Y., Xiao, L., Tian, J., Wang, X., Zhang, X., Fang, Y., & Li, W. (2022). Structural Characterization, Rheological Properties and Protection of Oxidative Damage of an Exopolysaccharide from Leuconostoc citreum 1.2461 Fermented in Soybean Whey. Foods, 11(15), 2283. for UV-vis analysis and FT-IR analysis.
4-Page 2 line 80, Please add the PCR program of 16S rRNA.
5- In the FT-IR analysis test for solids or semi-solids a range of 4000-400 cm-1 is used, why did you use up to 500 cm-1?
6-Page 4 line 165, The author indicates that he used a temperature range of 25–350 C. The results show that the compound decomposed at a higher degree than that used in the experiment. There is a contradiction between the working methods and the results.
7-Conclusions contain some results that should be deleted and rewritten to be more appropriate with the manuscript.
Author Response
Reviewer #2 comments:
Dear Editors and authors, The manuscript (Characterization and immunological activity of exopolysaccharide from Lacticaseibacillus paracasei GL1 isolated from Tibetan kefir grains) has a good idea but needs some correcting before it can be accepted for publication.
Response: Many thanks for your valuable comments for our manuscript. We have revised the manuscript to meet the publishing requirements.
1-The purpose of the manuscript is not clear, a clear aim should be written at the end of the introduction.
Response: Thank you for your valuable comment. We have added a clear aim of “The aim of this study was to research the structure and immune activity of the EPS of L. paracasei isolated from Tibetan kefir grains, so as to explore the relationship between polysaccharide activity and structure.” in Lines 66–68.
2-Scientific names should be reviewed and written according to the modern nomenclature of lactic acid bacteria.
Response: Thank you for your valuable comment. We have written the scientific names according to the modern nomenclature of lactic acid bacteria (LAB). We have written the LAB of different genera, and for the first occurrence of LAB, the genus name was in italics, the genus name was abbreviated, and the species name was full.
3-Many working methods do not contain scientific references such as Determination of pH and bacterial growth, UV-vis analysis and FT-IR analysis, 1D and 2D NMR spectra analysis, SEM analysis, The viability of RAW 264.7 cells, Effect of GL1-EPS on NO production and Determination of TNF-α, IL-1β and iNOS production. I suggest you two reference (Kimoto-Nira, H., Ohashi, Y., Amamiya, M., Moriya, N., Ohmori, H., & Sekiyama, Y. (2020). Fermentation of onion (Allium cepa L.) peel by lactic acid bacteria for production of functional food. Journal of Food Measurement and Characterization, 14(1), 142-149.). Ultrasound treatment (low frequency) effects on probiotic bacteria growth in fermented milk. Future of Food: Journal on Food, Agriculture and Society, 7(2), Nr-103.)) for Determination of pH and bacterial growth and (Li, Y., Xiao, L., Tian, J., Wang, X., Zhang, X., Fang, Y., & Li, W. (2022). Structural Characterization, Rheological Properties and Protection of Oxidative Damage of an Exopolysaccharide from Leuconostoc citreum 1.2461 Fermented in Soybean Whey. Foods, 11(15), 2283. for UV-vis analysis and FT-IR analysis.
Response: Thank you for your valuable comment. We have added references to the methods in this paper. We added “Kimoto-Nira, H., Ohashi, Y., Amamiya, M., Moriya, N., Ohmori, H., & Sekiyama, Y. Fermentation of onion (Allium cepa L.) peel by lactic acid bacteria for production of functional food. J. Food Meas. Charact. 2020, 14, 142–149. 10.1007/s11694-019-00276-4; Niamah, A.K. Ultrasound treatment (low frequency) effects on probiotic bacteria growth in fermented milk. Future Food. 2019, 7, Nr–103, 10.17170/kobra-20190709592” for the determination of pH and bacterial growth in Line 101. We added “Li, Y.Y., Xiao, L.Y., Tian, J.J. Wang, X.M., Zhang, X.L., Fang, Y., & Li, W. Structural characterization, rheological properties and protection of oxidative damage of an exopolysaccharide from Leuconostoc citreum 1.2461 fermented in soybean whey. Foods. 2022, 11, 2283. 10.3390/foods11152283” for the UV-vis analysis and FT-IR analysis in Line 121. We added “29. Wang, X.M. Xu, M.J., Xu, D.L., Ma, K., Zhang, C.L., Wang, G.X., Dong, M.S., & Li, W. Structural and prebiotic activity analysis of the polysaccharide produced by Lactobacillus helveticus SNA12. Carbohyd. Polym. 2022, 296, 119971. 10.1016/j.carbpol.2022.119971” for the detection of 1D and 2D NMR spectra in Line 153. We added “Suvakanta, D., Narsimha, M.P., Pulak, D., Joshabir, C., & Biswajit, D. Optimization and characterization of purified polysaccharide from Musa sapientum L. as a pharmaceutical excipient. Food Chem. 2014, 149, 76–83. 10.1016/j.foodchem.2013.10.068” for the detection of SEM in Line 158. We added “Han, C., Yang, J.K., Song, P.Y., Wang, X., & Shi, W.Y. Effects of salvia miltiorrhiza polysaccharides on lipopolysaccharide-induced inflammatory factor release in RAW264.7 cells. J. Interf. Cytok. Res. 2018, 1, 29–37. 10.1089/jir.2017.0087” for the viability detection of RAW 264.7 cells in Line 166. We added “Tian, J.J., Zhang, C.P., Wang, X.M., Rui, X., Zhang, Q.Q., Chen, X.H., Dong, M.S., & Li W. Structural characterization and immunomodulatory activity of intracellular polysaccharide from the mycelium of Paecilomyces cicadae TJJ1213. Food. Res. Int. 2021, 14, 71–13. 10.1016/j.foodres.2021.110515” for the detection of NO production in Line 188. We added “Shen, T., Wang, G.C., You, L., Zhang, L., Ren, H.W., Hu, W.C., Qiang, Q., Wang, X.F., Jia, L.L., Gu, Z.Z., & Zhao, X.X. Polysaccharide from wheat bran induces cytokine expression via the toll-like receptor 4-mediated p38 MAPK signaling pathway and prevents cyclophosphamide-induced immunosuppression in mice. Food Nutr. Res. 2017, 61, UNSP 1344523. 10.1080/16546628.2017.1344523” for the determination of TNF-α, IL-1β, and iNOS production in Line 191.
4-Page 2 line 80, Please add the PCR program of 16S rRNA.
Response: Thank you for your valuable suggestion. We added the PCR program of 16S rRNA of “The amplification was performed in 50 μL (final volume) of reaction mixture containing 25 μL Rapid Taq Master Mix, 2 uL each primer, 8 to 10 ng of purified genomic DNA 2 μL, and 19 μL deionized water. PCR parameters were 94 °C for 2 min, 30 cycles of 94 °C for 30 s, 55 °C for 30 s, and 72 °C for 90 s, and a final extension at 72 °C for 10 min.” in Lines 81–85.
5- In the FT-IR analysis test for solids or semi-solids a range of 4000-400 cm-1 is used, why did you use up to 500 cm-1?
Response: We are very sorry for our writing mistakes. FT-IR spectral analysis of polysaccharides was carried out in the range of 4000–400 cm−1. And we revised the sentence of “The FT-IR spectra were carried out on the Bruker Tensor-27 FT-IR spectrophotometer (Bruker Corp., Billerica, MA, USA) in the frequency range of 4000–400 cm-1 [26]” in Lines 119–121.
6-Page 4 line 165, The author indicates that he used a temperature range of 25–350 C. The results show that the compound decomposed at a higher degree than that used in the experiment. There is a contradiction between the working methods and the results.
Response: We are very sorry for our writing mistakes. During the analysis, the temperature range was set to 25–550 °C with a 10 °C /min heating rate. We revised the sentence of “During the analysis, the temperature range was set to 25–550 °C with a 10 °C /min heating rate.” in Lines 162–163.
7-Conclusions contain some results that should be deleted and rewritten to be more appropriate
Response: Thank you for your valuable suggestion. We deleted the results of “The average Mw of GL1-E1 and GL1-E2 were 3.9 × 105 Da and 8.2 × 105 Da, respectively, and they had mannose, glucose, and galactose with different molar ratios.”. And we revised conclusions into “They were heteropolysaccharides with different Mws and composed of mannose, glucose, and galactose with different molar ratios.” in Lines 513–514.

Round 2
Reviewer 1 Report
No comment